# Episodic Exploration for Deep Deterministic Policies for StarCraft Micromanagement

**Nicolas Usunier**[*], **Gabriel Synnaeve**[*], **Zeming Lin, Soumith Chintala**
Facebook AI Research
{usunier,gab,zlin,soumith}@fb.com

## Abstract

We consider scenarios from the real-time strategy game StarCraft as benchmarks for reinforcement learning algorithms. We focus on *micromanagement*, that is, the short-term, low-level control of team members during a battle. We propose several scenarios that are challenging for reinforcement learning algorithms because the state- action space is very large, and there is no obvious feature representation for the value functions. We describe our approach to tackle the micromanagement scenarios with deep neural network controllers from raw state features given by the game engine. We also present a heuristic reinforcement learning algorithm which combines direct exploration in the policy space and backpropagation. This algorithm collects traces for learning using deterministic policies, which appears much more efficient than, e.g., $\epsilon$-greedy exploration. Experiments show that this algorithm allows to successfully learn non-trivial strategies for scenarios with armies of up to 15 agents, where both Q-learning and REINFORCE struggle.

## 1 Introduction

StarCraft[1] is a real-time strategy (RTS) game in which each player must build an army and control individual units to destroy the opponent's army. As of today, StarCraft is considered one of the most difficult games for computers, and the best bots only reach the level of high amateur human players (Churchill, 2015). The main difficulty comes from the need to control a large number of units in partially observable environment, with very large state and action spaces: for example, in a typical game, there are at least $10^{1685}$ possible states whereas the game of Go has about $10^{170}$ states. Because of simultaneous and durative actions, StarCraft provides an ideal environment to study the control of many agents at large scale, and an opportunity to define tasks of increasing difficulty, from *micromanagement*, which concerns the short-term, low-level control of fighting units during battles, to long-term strategic and hierarchical planning under uncertainty. While building a controller for the full game based on machine learning is out-of-reach with current methods, we propose, as a first step, to study reinforcement learning (RL) algorithms in micromanagement scenarios in StarCraft.

Both the work on Atari games (Mnih et al., 2013) and the recent Minecraft scenarios studied by researchers (Abel et al., 2016; Oh et al., 2016) focus on the control of a single agent, with a fixed, limited set of actions. Coherently controlling multiple agents (units) is the main challenge of reinforcement learning for micromanagement tasks. This comes with two main challenges. The first one is to efficiently explore the large action space. The implementation of a coherent strategy requires the units to take actions that depend on each other, but it also implies that any small alteration of a strategy must be maintained for a sufficiently long time to properly evaluate the long-term effect of that change. In contrast to this requirement of consistency in exploration, the reinforcement learning algorithms that have been successful in training deep neural network policies such as Q-learning (Watkins & Dayan, 1992; Sutton & Barto, 1998) and REINFORCE (Williams, 1992; Deisenroth et al., 2013), perform exploration by randomizing actions. In the case of micromanagement, randomizing actions mainly disorganizes the units, which then rapidly lose the battle without collecting relevant

---

*: These authors contributed equally to this work.
[1]StarCraft and its expansion StarCraft: Brood War are trademarks of Blizzard Entertainment[TM]

feedback. The second challenge of micromanagement is that there is no obvious way to parameterize the policy given the state and the actions, because actions are relations between entities of the state, e.g. (unit A, attack, unit B) or (unit A, move, position B) and are not restricted to a few constant symbols such as "move left" or "move right". Multi-class architectures, such as these used for Atari games (Mnih et al., 2015), cannot evaluate actions that are parameterized by an entity of the state.

The contribution of this paper is twofold. First, we propose several micromanagement tasks from StarCraft (Section 3), then we describe our approach to tackle them and evaluate well known reinforcement learning algorithms on these tasks (Section 4). In particular, we present an approach of greedy inference to break out the complexity of taking the actions at each step. We also describe the features used to jointly represent states and actions, as well as a deep neural network model for the policy (Section 5). Second, we propose the zero order (ZO) reinforcement learning algorithm to address the difficulty of exploration in these tasks (Section 6). Compared to algorithms for efficient direct exploration in parameter space, the novelty of our algorithm is to explore directly in policy space by mixing parameter randomization and plain gradient descent.

## 2 RELATED WORK

Multi-agent reinforcement learning has been an active area of research (Busoniu et al., 2008). Most of the focus has been on learning agents in competitive environments with adaptive adversaries (Littman, 1994; Hu & Wellman, 1998; Tesauro, 2003). Some work has looked at learning control policies for individual agents in a collaborative setting with communication constraints (Tan, 1993; Bernstein et al., 2002), with applications such as soccer robot control (Stone & Veloso, 1999), and methods such as hierarchical reinforcement learning for communicating high-level goals (Ghavamzadeh et al., 2006), or learning an efficient communication protocol (Sukhbaatar et al., 2016). While the decentralized control framework is most likely relevant for playing full games of StarCraft, here we avoid the difficulty of imperfect information, therefore we use the multi-agent structure only as a means to structure the action space. As in the approach of (Maes et al., 2009) with reinforcement learning for structured output prediction, we use a greedy sequential inference scheme at each time frame: each unit decides on its action based solely on the state combined with the actions of units that came before it in the sequence.

Algorithms that have been used to train deep neural network controllers in reinforcement learning include Q-learning (Watkins & Dayan, 1992; Mnih et al., 2015), the method of temporal differences (Sutton, 1988; Tesauro, 1995), policy gradient and their variants (Williams, 1992; Deisenroth et al., 2013), and actor/critic architectures (Barto et al., 1983; Silver et al., 2014; 2016). Except for the deterministic policy gradient (DPG) (Silver et al., 2014), these algorithms rely on randomizing the actions at each step for exploration. DPG collects traces by following deterministic policies that remain constant throughout an episode, but can only be applied when the action space is continuous. Hausknecht & Stone (2015) apply DPG with paramterized action spaces, in which discrete actions (e.g. "move") are parameterized by continuous variables (e.g. the target location). Our work is most closely related to works that explore the parameter space of policies rather than the action space. Several approaches have been proposed that randomize the parameters of the policy at the beginning of an episode and run a deterministic policy throughout the entire episode, borrowing ideas from gradient-free optimization, e.g. (Mannor et al., 2003; Sehnke et al., 2008; Szita & Lörincz, 2006). However, these algorithms rely on gradient-free optimization for all parameters, which does not scale well with the number of parameters. Osband et al. (2016b) describe another type of algorithm where the parameters of a deterministic policy are randomized at the beginning of an episode, and learn a posterior distribution over the parameters as in Thomson sampling (Thompson, 1933). Their approach was proved to be efficient, but applies only to linear functions and scales quadratically with the number of parameters. The bootstrapped deep Q-networks (BDQN) (Osband et al., 2016a) are a practical implementation of the ideas of (Osband et al., 2016b) for deep neural networks. However, BDQN still performs exploration in the action space at the beginning of the training, and there is no randomization of the parameters. BDQN keeps several versions of the last layer of the deep neural network, and selects a single version per episode to perform Q-learning updates, while it ensembles all such "heads" as test time. In contrast, we randomize the parameters of the last layer once at the beginning of each episode and do not rely of estimates of a state-action value function.

In the context of RTS micromanagement, a large spectrum of AI approaches have been studied. There has been work on Bayesian fusion of hand-designed influence maps (Synnaeve & Bessiere, 2011), fast heuristic search in a simplified simulator (Churchill et al., 2012), and even evolutionary optimization (Liu et al., 2014). Overmind (Klein et al., 2010) used threat-aware A* pathing and RL-tuned potential fields. Closer to this work, Marthi et al. (2005) employ concurrent hierarchical Q-learning (units Q-functions are combined at the group level), Wender & Watson (2012) successfully applied tabular Q-learning (Watkins & Dayan, 1992) and SARSA (Sutton & Barto, 1998), with and without experience replay ("eligibility traces"), with a reward similar to the one used in several of our experiments. However, the action space was reduced to pre-computed "meta-actions": fight and retreat, and the features were hand-crafted. None of these approaches are used *as is* in existing StarCraft bots, for a lack of robustness, completeness (both can be attributed to hand-crafting), or computational efficiency. For a more detailed overview of AI research on StarCraft, the reader should consult (Ontanón et al., 2013).

## 3    STARCRAFT MICROMANAGEMENT SCENARIOS

We focus on micromanagement, which consists of optimizing each unit's actions during a battle. The tasks presented in this paper represent only a subset of the complexity of playing StarCraft. As StarCraft is a real-time strategy (RTS) game, actions are durative (are not fully executed on the next frame), and there are approximately 24 frames per second. As we take an action for each unit every few frames (e.g. every 9 frames here, more details can be found in Appendix D), we only consider actions that can be executed in this time frame, which are: the 8 move directions, holding the current position, an attack action for each of the existing enemy units. During training, we always control all units from one side, and the opponent (built-in AI in the experiments) is attacking us:

- m5v5 is a task in which we control 5 Marines (ranged ground unit), against 5 opponent Marines. A good strategy here is to focus fire, e.g. order all Marines to attack a single opponent.

- m15v16: same as above, except we have 15 Marines and the opponent has 16. A good strategy here is also to focus fire, while avoiding "overkill." 7 Marines attacking simultaneously kills an opponent in a single volley, so using more marines to simultaneously target an enemy causes attacks to be wasted, resulting in "overkill."

- dragoons_zealots: symmetric armies with two types of units: 3 Zealots (melee ground unit) and 2 Dragoons (ranged ground unit). Here a strategy requires to focus fire, and if possible to 1) not spend too much time having the Zealots walk instead of fight, 2) focus the Dragoons, who die more easily but deal more damage.

- w15v17: we control 15 Wraiths (ranged flying unit) while the opponent has 17. Flying units have no "collision", so multiple units can occupy the same tile and reach their target more quickly. It only takes 6 wraiths to kill an opponent in a single volley. Hence, it is important not to "overkill" on this map.

- other mXvY or wXvY scenarios. The 4 scenarios above are the ones on which we train our models, but they can learn strategies that overfit a given number of units, so we have similar scenarios but with different numbers of units (on each side).

For all these scenarios, a human expert can win 100% of the time against the built-in AI, by moving away units that are hurt (thus conserving firepower) and with proper focus firing.

## 4    FRAMEWORK: RL AND MULTIPLE UNITS

**Formalism**    The environment is approximated as a Markov Decision process (MDP), with a finite set of states denoted by $\mathcal{S}$. Each state $s$ has a set of units $\mathcal{U}(s)$, and a policy has to issue a command $c \in \mathcal{C}$ to each of them. The set of commands is finite. An *action* in that MDP is represented as a sequence of (unit, *command*) pairs $a = ((u_1, c_1), ..., (u_{|s|}, c_{|s|}))$ such that $\{u_1, ..., u_{|s|}\} = \mathcal{U}(s)$. $|s|$ denotes the number of units in state $s$ and $\mathcal{A}(s) = (\mathcal{U}(s) \times \mathcal{C})^{|s|}$ the set of actions in state $s$. We denote by $\rho(s'|s, a)$ the transition probability of the MDP and by $\rho_1$ the probability distribution of initial states. When there is a transition from state $s^t$ to a state $s^{t+1}$, the agent receives the reward $r^{t+1} = r(s^t, s^{t+1})$, where $r : \mathcal{S} \times \mathcal{S} \to \mathbb{R}$ is the reward function. We assume that commands are received and

executed concurrently, so that the order of commands in an action does not alter the transition probabilities. Finally, we consider the episodic reinforcement learning scenario, with finite horizon $T$ and undiscounted rewards. The learner has to learn a (stochastic) policy $\pi(a|s)$, which defines a probability distribution over actions in $\mathcal{A}(s)$ for every $s \in \mathcal{S}$. The objective is to maximize the expected undiscounted cumulative reward over episodes $R(\pi) = \mathbb{E}[\sum_{t=1}^{T-1} r(s^t, s^{t+1})] = \mathbb{E}[\bar{r}^{1..T}]$, where the expectation is taken with respect to $s^1 \sim \rho_1$, $s^{t+1} \sim \rho(.|a^t, s^t)$ and $a^t \sim \pi(.|s^t)$.

**The "greedy" MDP**   One way to break out the complexity of jointly inferring the commands to each individual unit is to perform greedy inference at each step: at each state, units choose a command one by one, knowing the commands that were previously taken by other units. Learning a greedy policy boils down to learning a policy in another MDP with fewer actions per state but exponentially more states, where the additional states correspond to the intermediate steps of the greedy inference. This reduction was previously proposed in the context of structured prediction by Maes et al. (2009), who proved that an optimal policy in this new MDP has the same cumulative reward as an optimal policy in the original MDP. We expand on this in Appendix B.

**Normalized cumulative rewards**   Immediate rewards are necessary to provide feedback that guides exploration. In the case of micromanagement, a natural reward signal is the difference between damage inflicted and incurred between two states. The cumulative reward over an episode is the total damage inflicted minus the total damage incurred along the episode. However, the scale of this quantity heavily depends on the number of units (both our units and enemy units, which significantly decreases along an episode) that are present in the state. Without proper normalization with respect to the number of units in the current state $z(s)$, learning will be artificially biased towards the large immediate rewards at the beginning of the episode. Then, instead of considering cumulative rewards from a starting state $s^t$, we define normalized cumulative rewards $\bar{n}^{t..T}$ as the following recursive computation over an episode:

$$\forall t \in \{1, ..., T-1\}, \bar{n}^{t..T} = \frac{r^{t+1} + z(s^{t+1})\bar{n}^{t+1..T}}{z(s^t)} \ . \tag{1}$$

We use the sum of maximum hit points of all units in the state $s^t$ as normalization factor $z(s^t)$, which implies that $\bar{n}^{t..T} \in [-0.5, 0.5]$. One way to look at this normalization process is to consider that the reward is $\frac{r^{t+1}}{z(s^t)}$, and $\frac{z(s^{t+1})}{z(s^t)}$ plays the role of an (adaptive) discount factor, which is chosen to be at most 1, and strictly smaller than 1 when the number of units change.

For policy gradient and our algorithm described in section 6, we directly use $\bar{n}^{t..T}$. We describe in Appendix C how we adapted the update rule for Q-learning.

## 5   FEATURES AND MODEL FOR MICROMANAGEMENT IN STARCRAFT

We describe in this section the features and the neural network architecture we use to parameterize the policy. Since we consider the greedy inference described in the previous section, the underlying MDP will contain states of the form $\tilde{s} = (s, a_{1..k}, u_{k+1})$, where: $s$ is the current state of the game given by the game engine, $k$ is the number of units which already "played" at this frame, $a_{1..k}$ is the sequence of the $k$ pairs $(unit, command)$ that correspond to the $k$ commands the have already been chosen, and finally $u_{k+1}$ is the unit to play. For each unit, we consider two types of commands: (1) attack a given enemy unit, and (2) move to a specific position. In order to reduce the number of possible move commands, we only consider 9 move commands, which either correspond to a move in one of the 8 basic directions, or staying at the same position.

There are several challenges to represent states and actions in RTS games:

- The number of units and actions are not bound a priori and varies in time
- Commands must be evaluated in context of all currently executing commands
- Attack actions must resolve the reference to its target

To address the first two challenges, we adopt an approach based on a joint encoding of states and commands. Denoting by $\tilde{s} = (s, a_{1..k}, u_{k+1})$ the current state of the greedy MDP and $c$ a

Table 1: Unit features as given by the game engine, their abbreviated name and their type: *cat.* means the feature is caterogical and 1-hot encoded, real-valued features comme with the re-scaling constant.

| hit points | shield | cooldown | is enemy | unit type |
|------------|--------|----------|----------|-----------|
| ($hp, \in \mathbb{R}$, /20) | ($shield, \in \mathbb{R}$, /20) | ($cd, \in \mathbb{R}$, /10) | ($nmy, bool$) | ($type, cat.$) |
| position | previous target | chosen target | prev. cmd type | chosen cmd type |
| ($pos, \in \mathbb{R}^2$, /20) | ($tgt\_pos, \in \mathbb{R}^2$, /20) | ($next\_pos, \in \mathbb{R}^2$, /20) | ($prev\_cmd, cat.$) | ($next\_cmd, cat.$) |

candidate action, we learn the parameters $w$ and $\theta$ of a (state, command) value function of the form $f(\tilde{s}, c) = \langle w, \Psi_\theta(\tilde{s}, c) \rangle$ where $w \in \mathbb{R}^d$ and $\Psi_\theta(\tilde{s}, c)$ is the output of a embedding network that maps (state, command) pairs to $\mathbb{R}^d$, with parameters $\theta$. In Q-learning and our algorithm presented in the next section, we directly use $f$ as the state/action value function, whereas in policy gradient the probability to take command $c$ in state $\tilde{s}$ is given by the Gibbs distribution over $f(\tilde{s}, c)$ with temperature $\tau$: $\pi(c|\tilde{s}) = \frac{e^{f(\tilde{s}, c)/\tau}}{\sum_{c' \in C} e^{f(\tilde{s}, c')/\tau}}$.

To tackle the last challenge, we identify units with their $(x, y)$ coordinates in the map. We add two fields to the unit features that contain the coordinates of their corresponding target, or its own location if it does not have a target. To evaluate a command $c = (\text{<actor unit>}, \text{<attack or move>}, \text{<target>})$, we compute pairwise distances between the actor and the target. Note that with this kind of representation, the input of the embedding network $\Psi_\theta$ is a joint representation of the state $\tilde{s}$ and the command $c$ to evaluate. A complete list of unit features is given in Table 1. *Hit points* are the remaining life points of the unit, *shield* corresponds to additional hit points that are not affected by armor and regenerate slowly, *cooldown* is the time to wait until damages can be inflicted.

The full scoring approach is depicted in Figure 1. In our approach, a state is represented as a list of units. The raw features are transformed by a featurizer that 1) takes the 3 unit features (*pos*, *tgt_pos* and *next_pos*) and computes their distances with the position the acting unit and its target ($pos_c$ and $tgt_c$). All 4 categorical variables are passed through a 10-dimensional linear embedding (not shown in figure). In addition to the 4 real valued unit features, we have a 40 dimensional feature vector per unit as input to our network.

Each unit feature vector then goes through the unit-level embedding network. We then concatenate the max and mean poolings across units with an embedding of the command type. Then, the resultant 210 dimensional vector is passed through a final state-command embedding network. Both the unit-level and state-command embedding networks have a hidden dimension of 100, and ELU nonlinearities in the intermediate layer (Clevert et al., 2015). We use tanh for the final unit-level network nonlinearty, and a ReLU for the final state-command network nonlinearity. We did not extensively experiment with the structure of the network, but we found the maxpooling and tanh nonlinearity to be particularly important.

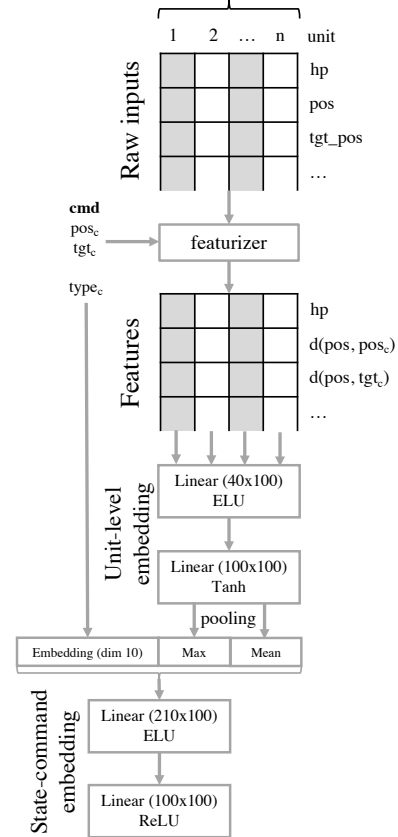

Figure 1: Representation of the joint (state, command) featurization and scoring process.

The advantage of this approach is to rely on raw features only, and does not require any encoding of the game dynamics, in contrast to previous works on RL for micromanagement (see e.g. (Wender & Watson, 2012)) that used domain knowledge handcrafted in the features (such as the damages inflicted by an attack). The distance-based encoding is also a simple way to represent the different relationships between units that correspond to previous/chosen attacks.

# 6  COMBINING BACKPROPAGATION AND ZERO-ORDER OPTIMIZATION

Our preliminary experiments with Q-learning or REINFORCE made it clear that structured exploration was necessary to learn non-trivial strategies with substantial armies. The randomization of actions lead to the disorganization of the army and a rapid defeat, which prevents the algorithms from evaluating alterations to the current policy in the long run. Whereas gradient-free optimization that performs episode-based exploration (e.g. Mannor et al. (2003); Sehnke et al. (2010)) would be a valid choice, it only scales to few parameters. Preliminary experiments with direct exploration in the parameter space of the deep neural network confirmed that a more efficient scheme was needed.

The deterministic policy we consider takes action $a$ in state $s$ according to the rule

$$\pi_{w,\theta}(s) = \operatorname*{argmax}_{a \in \mathcal{A}(s)} \langle w, \Psi_\theta(s,a) \rangle \, .$$

We use the notation $(s, a)$ for state and actions in an MDP for the presentation of the algorithm, even though in our experiments we use it with states $\tilde{s}$ of the greedy MDP and unit-level commands $c$. Likewise, we describe the algorithm in the standard cumulative reward setup, while in our experiments we use the normalized cumulative rewards.

This form of policy naturally allows to perform structured exploration by only randomizing parts of the network. More specifically, the parameters $w$ of the last layer affect all states and actions in a similar way along an episode. The approach we follow is then to perform gradient-free optimization on these parameters $w$ only. Following stochastic methods for zero-th order optimization (Kiefer et al., 1952; Nemirovsky et al., 1982; Spall, 1997; Duchi et al., 2013; Ghadimi & Lan, 2013), the gradient of a differentiable function $x \in \mathbb{R}^d \mapsto f(x)$ can be estimated by

$$\nabla f(x) \approx \mathbb{E}[\frac{d}{\delta} f(x + \delta u) u] \, ,$$

where the expectation is taken over the vector $u$ sampled on the unit sphere (Nemirovsky et al., 1982, chapter 9.3). The constant $\frac{d}{\delta}$ is absorbed by learning rates, so we ignore it in the following. Given a (state, action) pair $(s, a)$ and the observed cumulative reward $\bar{r}^{1..t}$ for an episode of length $t$, an estimate of the gradient of the expected cumulative reward with respect to $w$ is thus $\bar{r}^{1..t} u$. In practice, we use $\frac{\sum_{k=1}^{t-1} \bar{r}^{k..t}}{t} u$ rather than $\bar{r}^{1..t} u$, which corresponds to the gradient of the average cumulative reward over the episode. We did not observe a large difference in preliminary experiments.

The overall algorithm is described in Algorithm 1. At the beginning of an episode, a perturbation $u$ is sampled from the unit sphere of $\mathbb{R}^d$ and the policy $s \mapsto \pi_{w+\delta u,\theta}(s)$ is run through the entire episode ($\delta$ is a hyperparameter of the algorithm). The perturbation vector plays both role of performing structured exploration and providing the gradient estimate of the cumulative reward with respect to $w$. The algorithm performs a minibatch update at the end of the episode. The second loop in Algorithm 1 accumulates the update direction for $w$ in $\hat{g}(w)$ and the update direction for $\theta$ in $\hat{G}(\Theta)$. The update (*) ensures that $\hat{g}(w) = \frac{\sum_{k=1}^{t-1} \bar{r}^{k..t}}{t} u$ at the end of the loop, as described in the previous paragraph.

The deterministic exploration along an episode does not provide any update rule for the parameters of the embedding network, because the randomization is the same for every (state, action) pair. We propose a heuristic rule to update the parameters $\theta$ of the embedding network, motivated by the following remark: given a function $(w \in \mathbb{R}^d, v \in \mathbb{R}^d) \mapsto F(\langle w, v \rangle) \in \mathbb{R}$, we have $\nabla_w F = F'(\langle w, v \rangle) v$ and $\nabla_v F = F'(\langle w, v \rangle) w$. Denoting by $\frac{w}{v}$ the term-by-term division of vectors (assuming $v$ contains only non-zero values) and $\odot$ the term-by-term multiplication operator, we obtain:

$$\nabla_v F = (\nabla_w F) \odot \frac{w}{v} \, .$$

Taking $F$ to be the cumulative reward in (state, action) pair $(s^k, a^k)$, assuming that it only depends on the dot product $\langle w, \Psi_\theta(s^k, a^k) \rangle$, the update rule above gives $u \bar{r}^{k..t} \odot \frac{w}{\Psi_\theta(s,a)}$ as input for backpropagation. In practice, we use the sign of $\frac{w}{\Psi_\theta(s,a)}$ to avoid exploding gradients due to the division by $\Psi_\theta(s,a)$, which seemed to work as well as more involved clipping heuristics. Line (**) of Algorithm 1 accumulates along the episode the updates for $\theta$ in $\hat{G}(\theta)$. In (**), we divide the cumulative reward by $t$ to optimize the average reward over the episode, and $z \mapsto \text{backprop}_{\Psi_\theta(s^k, a^k)}(z)$ refers to the gradient with respect to $\theta$ when the network input is $(s^k, a^k)$ and the backward step uses $z$ as input.

**Input** exploration hyper-parameter $\delta$, learning rate $\eta$,
(state, action) embedding network $\Psi_\theta(s, a)$ taking values in $\mathbb{R}^d$, with parameters $\theta \in \mathbb{R}^m$.;
**initialization**: $w \leftarrow 0$, $w \in \mathbb{R}^d$;
**while** *stopping criterion not met* **do**

 Sample $u$ uniformly on the unit sphere of $\mathbb{R}^d$;
 `// follow the perturbated deterministic policy for one episode`
 $t \leftarrow 0$;
 **while** *episode not ended* **do**

 $t \leftarrow t + 1$;
 observe current state $s^t$ and reward $r^t$;
 choose action $a^t = \mathrm{argmax}_{a \in \mathcal{A}(s)} \langle w + \delta.u, \Psi_\theta(s^t, a) \rangle$;

 **end**
 $\hat{g}(w) = 0$ `// accumulation of minibatch update for w`
 $\hat{G}(\theta) = 0 \in \mathbb{R}^{m \times d}$ `// accumulation of minibatch update for` $\theta$
 $\bar{r}^{t..t} = 0$ `// cumulative reward, use` $\bar{n}^{t..t} = 0$ `for normalized rewards`
 **for** $k = t - 1$ **to** $1$ **do**

 $\bar{r}^{k..t} = \bar{r}^{k+1..t} + r^{k+1}$ `// for normalized rewards, use` $\bar{n}^{k..t}$ `and (1)`
 $\hat{g}(w) \leftarrow \hat{g}(w) + \frac{\bar{r}^{k..t}}{t} u$ (*);
 $\hat{G}(\theta) \leftarrow \hat{G}(\theta) + \mathrm{backprop}_{\Psi_\theta(s^k, a^k)} \left( \frac{\bar{r}^{k..t}}{t} u \odot \left( sign \frac{w}{\Psi_\theta(s^k, a^k)} \right) \right)$ (**);

 **end**
 `// perform gradient ascent`
 update_adagrad($w$, $\eta \hat{g}(w)$);
 update_adagrad($\theta$, $\eta \hat{G}(\theta)$);

**end**

**Algorithm 1:** Zero-order (ZO) backpropagation algorithm

The reasoning above is only an intuitive motivation of the update rule (**) of Algorithm 1, because we neglected that a single $u$ is sampled for an entire episode. We also neglected the $\mathrm{argmax}$ operation that chooses the actions. Nonetheless, considering (**) as a crude approximation to some real estimator of the gradient seems to work very well in practice, as we shall see in our experiments. Finally, we use Adagrad (Duchi et al., 2011) to update the parameters of the different layers. We found the use of Adagrad's update scheme fairly important in practice, compared to other approaches such as e.g. RMSProp (Tieleman & Hinton, 2012), even though RMSProp tended to work slightly better with Q-learning or REINFORCE in our experiments.

# 7 EXPERIMENTS

## 7.1 SETUP

We use Torch7 (Collobert et al., 2011) for all our experiments. We connect our Torch code and models to StarCraft through a socket server, as described in (Synnaeve et al., 2016). We ran experiments with deep Q networks (DQN) (Mnih et al., 2013), policy gradient (PG) (Williams, 1992) (detailed in Appendix A), and zero order (ZO). We did an extensive hyper-parameters search, in particular over $\epsilon$ (for epsilon-greedy exploration in DQN), $\tau$ (for policy gradient's softmax), learning rates, optimization methods, RL algorithms variants, and potential annealings (detailed Appendix E).

## 7.2 BASELINE HEURISTICS

As all the results that we report are against the built-in AI, we compare our win rates to the ones of baseline heuristics. Some of these heuristics often perform the micromanagement in full-fledged StarCraft bots (Ontanón et al., 2013), and are the basis of heuristic search (Churchill et al., 2012). The baselines are the following:

Figure 2: Example of the training uncertainty (one standard deviation) on 5 different initialization for DQN (left) and zero-order (right) on the m5v5 scenario.

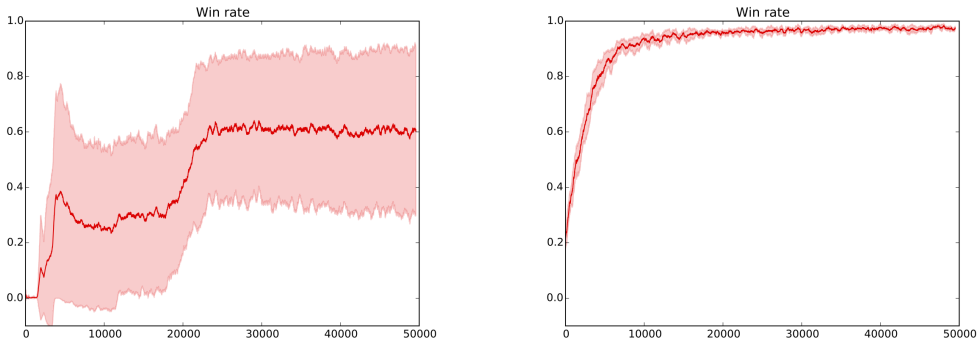

- *random no change* (rand_nc): select a random target for each of our units and do not change this target before it dies (or our unit dies). This spreads damage over several enemy units, but when there are collisions, it may make our units to move a lot to be in range of their target.

- *noop*: send no action. In this case, the built-in AI will control our units, so this exhibit the symmetry (or not!) of a given scenario. As we are always in a defensive position, with the enemy commanded to walk towards us, all other things considered equal, it should be easier for the defending built-in AI than for the attacking one. Our models cannot send a noop command.

- *closest* (c): each of our units targets the enemy unit closest to it. This is not a bad heuristic as enemy units formation will make it so that several of our units have the same opponent unit as closest unit (some form of focus firing), but not all of them (no overkill). It is also quite robust for melee units (e.g. Zealots) as it means they spend less time moving and more time attacking.

- *weakest closest* (wc): each of our units targets the weakest enemy unit. The distance of the enemy unit to the center of mass of our units is used for tie-breaking. This may overkill.

- *no overkill no change* (nok_nc): same as the weakest closest heuristic, but register the number of our units that target each opponent unit, choosing another target to focus fire when it becomes overkill to keep targeting a given unit. Each of our units keep firing on their target without changing (that would lead to erratic behavior). Our implementation of the "no overkill" component does not take all the dynamics of the game into account, and so if our units die without doing their expected damage on their target, "no overkill" can be detrimental.

## 7.3 RESULTS

The first thing that we looked at were sliding average win rates over 400 battles during training against the built-in AI of the various models. In Figure 2, we can see than DQN is much more dependent on initialization and variable than zero order (ZO). DQN can unlearn, reach suboptimal plateau, or overall need a lot of exploration to start learning (high sample complexity).

For all the results that we present in Tables 2 and 3, we ran the models in "test mode" by making them deterministic. For DQN we remove the epsilon-greedy exploration (set $\epsilon = 0$), for PG we do not sample from the Gibbs policy but instead take the value-maximizing action, and for ZO we do not add noise to the last layer.

We can see in Table 2 that m15v16 is at the advantage of our player's side (*noop* is at 81% win rate), whereas w15v17 is hard (*c* is at 20% win rate). By looking just at the results of the heuristics, we can see that overkill is a problem on m15v16 and w15v17 (*nok_nc* is better than *wc*). "Attack closest" (*c*) is approximatively as good as *nok_nc* at spreading damage, and thus better on m15v16 because there are lots of collisions (and attacking the closest unit is going to trigger less movements).

Overall, the zero order optimization outperforms both DQN and PG (REINFORCE) on most of the maps. The only map on which DQN and PG perform well is m5v5. It seems to be easier to learn

Table 2: Test win rates over 1000 battles for the training scenarios, for all methods and for heuristics baselines. The best result for a given map is in bold.

| | heuristics | | | | | RL | | |
|---|---|---|---|---|---|---|---|---|
| map | rand_nc | noop | c | wc | nok_nc | DQN | PG | ZO |
| dragoons_zealots | .14 | .49 | .67 | .83 | .50 | .61 | .69 | **.90** |
| m5v5 | .49 | .84 | .94 | .96 | .83 | **.99** | .92 | **1.** |
| m15v16 | .00 | **.81** | **.81** | .10 | .68 | .13 | .19 | .79 |
| w15v17 | .19 | .10 | .20 | .02 | .12 | .16 | .14 | **.49** |

Table 3: Win rates over 1000 games for out-of-training-domain maps, for all methods. The map on which this method was trained on is indicated on the left. The best result is in bold, the best result out of the reinforcement learning methods is in italics.

| train map | test map | best heuristic | DQN | PG | ZO | train map | test map | best heuristic | DQN | PG | ZO |
|---|---|---|---|---|---|---|---|---|---|---|---|
| m15v16 | m5v5 | **.96** (wc/c) | *.96* | .79 | .80 | w15v17 | w5v5 | **.78** (c) | .70 | .70 | *.74* |
| | m15v15 | **.97** (c) | .27 | .16 | *.80* | | w15v13 | 1. (rand_nc/c) | 1. | .99 | 1. |
| | m18v18 | **.98** (c/noop) | .18 | .25 | *.82* | | w15v15 | **.95** (c) | .87 | .61 | *.99* |
| | m18v20 | **.63** (noop) | .00 | .01 | *.17* | | w18v18 | **.99** (c) | .92 | .56 | *1.* |
| | | | | | | | w18v20 | .71 (c) | .31 | .24 | *.76* |

a focus firing heuristic (e.g. "attack weakest") by identifying and locking on a feature, than to also learn not to "overkill". We interpret the learned behaviors in Appendix F.

We then studied how well a model trained on one map performs on maps with a different number of units, to test generalization. Table 3 contains the results for this experiment. We observe that DQN performs the best on m5v5 when trained on m15v16, because it learned a simpler (but more efficient on m5v5) heuristic. "Noop" and "attack closest" are quite good with the large Marines map because they generate less moves (and less collisions). Overall, ZO is consistently significantly better than other RL algorithms on these generalization tasks, even though it does not reach an optimal strategy.

We also played the best model on each map against each other. We modify the maps in this case such that they are all symmetric, but with the same army composition. Table 4 shows the results for this experiment. It seems that PG and DQN learned very different strategies on wXvY, DQN beats PG consistently when trained on w15v17, while the PG model trained on w15v15 has an edge over DQN. Overall, ZO comes out ahead in every match-up except for m5v5, often by a significant margin.

## 8 CONCLUSION

This paper presents two main contributions. First, it establishes StarCraft micromanagement scenarios as complex benchmarks for reinforcement learning: with durative actions, delayed rewards, and large action spaces making random exploration infeasible. Second, it introduces a new reinforcement learning algorithm that performs better than prior work (DQN, PG) for discrete action spaces in these micromanagement scenarios, with robust training (see Figure 2) and episodically consistent exploration (exploring in the policy space).

This work leaves several doors open and calls for future work. Simpler embedding models of state and actions, and variants of the model presented here, have been tried, none of which produced efficient units movement (e.g. taking a unit out of the fight when its hit points are low). There is ongoing

Table 4: Win rates over 2000 games against each other.

| trained on | dragoons_zealots | m15v16 | | m5v5 | w15v15 | | w15v17 | |
|---|---|---|---|---|---|---|---|---|
| tested on | dragoons_zealots | m15v15 | m18v18 | m5v5 | w15v15 | w18v18 | w15v15 | w18v18 |
| PG > DQN | .74 | .46 | .47 | .49 | .61 | .69 | .09 | .04 |
| ZO > PG | .76 | .82 | .79 | .44 | .82 | .77 | .98 | .99 |
| ZO > DQN | .93 | .85 | .86 | .39 | .88 | .90 | .79 | .80 |

work on convolutional networks based models that conserve the 2D geometry of the game (while embedding the discrete components of the state and actions). The zero order optimization technique presented here should be studied more in depth, and empirically evaluated on domains other than StarCraft (e.g. Atari). As for StarCraft scenarios specifically, the subsequent experiments will include self-play in training, multi-map training (more generic models), and more complex scenarios which include several types of advanced units with actions other than move and attack. Finally, the goal of playing full games of StarCraft should not get lost, so future scenarios would also include the actions of "recruiting" units (deciding which types of unit to use), and how to best make use of them.

## ACKNOWLEDGEMENTS

We thank Y-Lan Boureau, Antoine Bordes, Florent Perronnin, Dave Churchill, Léon Bottou and Alexander Miller for helpful discussions and feedback about this work and earlier versions of the paper. We thank Timothée Lacroix and Alex Auvolat for technical contributions to our StarCraft/Torch bridge. We thank Davide Cavalca for his support on Windows virtual machines in our cluster environment.

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

## A    BASELINES

We here briefly describe the two algorithms we use as baseline, Q-learning (Sutton & Barto, 1998) and REINFORCE (Williams, 1992).

**Q-learning**    The Q-learning algorithm in the finite-horizon setting learns an action-value function $Q$ by solving the Bellman equation

$$\forall s \in \mathcal{S}, \forall a \in \mathcal{A}(s), Q_t(s, a) = \sum_{s' \in \mathcal{S}} \rho(s'|s, a)\big(r(s, s') + \max_{a' \in \mathcal{A}(s')} Q_{t+1}(s', a')\big), \qquad (2)$$

where $Q_t$ is the state-action value function at stage $t$ of an episode, and $Q_T(s, a) = 0$ by convention. $Q_t(s, a)$ is also $0$ whenever a terminal state is reached, and transitions from a terminal state only go to the same terminal state.

Training is usually carried out by collecting traces $(s^t, a^t, s^{t+1}, r^{t+1})_{t=1,...,T-1}$ using $\epsilon$-greedy exploration: at state $s$ and stage $t$, an action in $\mathrm{argmax}_{a \in \mathcal{A}(s)} Q_t(s, a)$ is chosen with probability $1-\epsilon$, or an action in $\mathcal{A}(s)$ is chosen uniformly at random with probability $\epsilon$. In practice, we use stationary $Q$ functions (i.e., $Q_t = Q_{t+1}$), which are neural networks, as described in Section 5. Training is carried out using the standard online update rule for Q learning with function approximation (see (Mnih et al., 2015) for DQN), which we apply in mini-batches (hyper-parameters are detailed in Appendix E).

This training phase is distinct from the test phase, in which we record the average cumulative reward of the deterministic policy[2] $s \mapsto \mathrm{argmax}_{a \in \mathcal{A}(s)} Q(s, a)$.

**REINFORCE**    The algorithm REINFORCE belongs to the family of policy gradient algorithms (Sutton et al., 1999). Given a stochastic policy $\pi_\Theta$ parameterized by $\Theta$, learning is carried out by generating traces $(s^t, a^t, s^{t+1}, r^{t+1})_{t=1,...,T-1}$ by following the current policy. Then, stochastic gradient updates are performed, using the gradient estimate:

$$\sum_{t=1}^{T} \bar{r}(s^{t..T}) \nabla_\Theta \log(\pi_\Theta(a^t|s^t)). \qquad (3)$$

We use a Gibbs policy (with temperature parameter $\tau$) as the stochastic policy:

$$\pi_\Theta(a|s) = \frac{\exp(\phi_\Theta(a, s)/\tau)}{\sum_{b \in \mathcal{A}(s)} \exp(\phi_\Theta(b, s)/\tau)},$$

where $\phi_\Theta$ is a neural network with paramters $\Theta$ that gives a real-valued score to each (state, action) pair. For testing, we use the deterministic policy $\pi_\Theta(s) = \mathrm{argmax}_{a \in \mathcal{A}(s)} \phi_\Theta(a, s)$.

## B    THE GREEDY MDP

A natural way to define the greedy MDP (Section 4) is to define the set of atomic actions of the greedy policy as all possible (unit, command) pairs for the units whose command is still not decided. This would lead to an inference with quadratic complexity with respect to the number of units, which is undesirable.

We settled on iteratively choosing a unit, then a command to apply to that unit, which yields an algorithm with $2|s|$ steps for state $s$, linear in the number of units. Since the commands are executed concurrently by the environment after all commands have been decided, the cumulative reward does not depend on the order in which we choose the units, for instance: uniformly at random among remaining units. More formally, using the notation $a_{1..k}$ to denote the $k$ first (unit, command) pairs of an action $a$ (with the convention $a_{1..0} = \emptyset$), the state space $\tilde{\mathcal{S}}$ of the greedy MDP is defined by

$$\tilde{\mathcal{S}} = \big\{(s, a_{1..k}, u_{k+1}) \mid s \in \mathcal{S}, 0 \le k < |s|, a = ((u_1, c_1), ..., (u_{|s|}, c_{|s|})) \in \mathcal{A}(s)\big\}.$$

---

[2]The policy may not be determistic if we break ties randomly in the argmax.

The action space $\mathcal{A}(\tilde{s})$ of each state $\tilde{s} \in \tilde{\mathcal{S}}$ is constant and equal to the set of commands $\mathcal{C}$. Moreover, for each state $s$ of the original MDP, any action $a = ((u_1, c_1), ..., (u_{|s|}, c_{|s|}) \in \mathcal{A}(s)$, the transition probabilities $\tilde{\rho}$ in the greedy MDP are defined by

$$\forall k \in \{0, ..., |s| - 1\}, \; \tilde{\rho}((s, a_{1..k}, u_{k+1}) | (s, a_{1..k-1}, u_k), c_k) = \frac{1}{|s| - k}$$

$$\text{and} \; \forall s' \in \mathcal{S}, \forall u' \in \mathcal{U}(s'), \; \tilde{\rho}((s', \emptyset, u') | (s, a_{1..|s|-1}, u_{|s|}), c_{|s|}) = \frac{1}{|s'|} \rho(s'|s, a).$$

Finally, using the same notation as above, the reward function $\tilde{r}$ between states that represent intermediate steps of the algorithm is $0$ and the last unit to play receives the reward:

$$\tilde{r}((s, a_{1..k-1}, u_k), (s, a_{1..k}, u_{k+1})) = 0, \; \text{and} \; \tilde{r}((s, a_{1..|s|-1}, u_{|s|}), (s', \emptyset, u')) = r(s, s').$$

It can be shown that an optimal policy for this greedy MDP chooses actions that are optimal for the original MDP, because the immediate reward in the original MDP does not depend on the order in which the actions are taken. This result only applies if the family of policies has enough capacity. In practice, some ordering may be easier to learn than others, but we did not investigate this issue because the gain, in terms of computation time, of the random ordering was critical for the experiments.

## C  NORMALIZED CUMULATIVE REWARDS

The normalized rewards (from Section 4) maintain the invariant $\bar{n}^{t..T} = \frac{\bar{r}^{t..T}}{z(s^t)}$; but more importantly, the normalization can be applied to the Bellman equation (2), which becomes

$$\forall s \in \mathcal{S}, \forall a \in \mathcal{A}(s), Q(s, a) = \sum_{s' \in \mathcal{S}} \frac{\rho(s'|s, a)}{z(s)} \left( r(s, s') + z(s') \max_{a' \in \mathcal{A}(s')} Q(s', a') \right).$$

This normalization does not change the optimal policy because it maintains the invariant that the expected normalized cumulative reward from a given state $s$ to the end of an episode (by following the optimal deterministic policy) is the expected cumulative reward from this $s$ divided by a value that depends only on $s$.

The stochastic gradient updates for Q-learning can easily be modified accordingly, as well as the gradient estimate in REINFORCE (3) in which we replace $\bar{r}$ by $\bar{n}$.

## D  STARCRAFT SPECIFICS

We advocate that using existing video games for RL experiments is interesting because the simulators are oftentimes complex, and we (the AI programmers) do not have control about the source code of the simulator. In RTS games like StarCraft, we do not have access to a simulator (and writing one would be a daunting task), so we cannot use (Monte Carlo) tree search (Gelly & Wang, 2006) directly, even less so in the setting of full games (Ontanón et al., 2013). In this paper, we consider the problem of micromanagement scenarios, a subset of full RTS play. Micromanagement is about making good use of a given set of units in an RTS game. Units have different features, like range, cooldown, hit points (health), attack power, move speed, collision box etc. These numerous features and the dynamics of the game advantage player that take the right actions at the right times. Specifically for the game(s) StarCraft, for which there are professional players, very good competitive players and professional players perform more than 300 actions per minute during intense battles.

We ran all our experiments on simple scenarios of battles of an RTS game: StarCraft: Broodwar. These scenarios can be considered small scale for StarCraft, but they already deem challenging for existing RL approaches. The joint action space is in $\Theta((\#\text{commands per unit})^{\#\text{units}})$, with a peak number of units of about $400$ (Synnaeve & Bessiere, 2011). For an example scenario of 15 units (that we control) against 16 enemy units, even while reducing the action space to "atomic" actions (surrounding moves, and attacks), we obtain 24 (8+16) possible discrete actions per unit for our controller to choose from ($24^{15}$ actions total) at the beginning of the battle. Battles last for tens of seconds, with durative actions, simultaneous moves, and at 24 frames per second. The strategies that we need to learn consist in coordinated sets of actions that may need to be repeated, e.g. focus firing without overkill. We use a featurization that gives access only to the state from the game, we do not

pre-process the state to make it easier to learn a given strategy, thus keeping the problem elegant and unbiased.

For most of these tasks ("maps"), the number of units that our RL agent has to consider changes over an episode (a battle), as do its number of actions. The fact that we are playing in this specific adversarial environment is that if the units do not follow a coherent strategy for a sufficient amount of time, they will suffer an unrecoverable loss, and the game will be in a state of the game where the units will die very rapidly and make little damage, independently of how they play – a state that is mostly useless for learning.

Our tasks ("maps") represent battles with homogeneous types of units, or with little diversity (2 types of unit for each of the players). For instance, they may use a unit of type Marine, that is one soldier with 40 hit points, an average move speed, an average range (approximately 10 times its collision size), 15 frames of cooldown, 6 of attack power of normal damage type (so a damage per second of 9.6 hit points per second, on a unit without armor). On symmetric and/or monotyped maps, strategies that are required to win (on average) are "focus firing", without overkill (not more units targeting a unit than what is needed to kill it). For perfect win rates, some maps may require that the AI moves its units out from the focus firing of the opponent.

## E  HYPER-PARAMETERS

Taking an action on every frame (24 times per second at the speed at which human play StarCraft) for every unit would spam the game needlessly, and it would actually prevent the units from moving[3]. We take actions for all units synchronously on the same frame, even `skip_frames` frames. We tried several values of this hyper-parameter (5, 7, 9, 11, 13, 17) and we only saw smooth changes in performance. We ran all the following experiments with a `skip_frames` of 9 (meaning that we take about 2.6 actions per unit per second).We also report the strongest numbers for the baselines over all these skip frames. We optimize all the models after each battle (episode), with RMSProp (momentum 0.99 or 0.95), except for zero-order for which we optimized with Adagrad (Adagrad did not seem to work better for DQN nor REINFORCE). In any case, the learning rate was chosen among $\{10^{-2}, 10^{-3}, 10^{-4}\}$.

For all methods, we tried experience replay, either with episodes (battles) as batches (of sizes 20, 50, 100), or additionally with random batches of $(s_t, a_t, r_{t+1}, s_{t+1}, terminal?)$ quintuplets in the case of Q-learning, it did not seem to help compared to batching with the last battle. So, for consistency, we only present results where the training batches consisted of the last episode (battle).

For Q-learning (DQN), we tried two schemes of annealing for epsilon greedy, $\epsilon = \frac{\epsilon_0}{\sqrt{1+\epsilon_a.\epsilon_0.t}}$ with $t$ the optimization batch, and $\epsilon = \max(0.01, \frac{\epsilon_0}{\epsilon_a.t})$, Both with $\epsilon_0 \in \{0.1, 1\}$, and respectively $\epsilon_a \in \{0, \epsilon_0\}$ and $\epsilon_a \in \{10^{-5}, 10^{-4}, 10^{-3}\}$. We found that the first works marginally better and used that in the subsequent experiments with $\epsilon_0 = 1$ and $\epsilon_a = 1$ for most of the scenarios. We also used Double DQN as in (Van Hasselt et al., 2015) (thus implemented as target DQN). For the target/double network, we used a lag of 100 optimizations, thus a lag of 100 battles in all the following experiments. According to our initial runs/sweep, it seems to slightly help for some cases of over-estimation of the Q value.

For REINFORCE we searched over $\tau \in \{0.1, 0.5, 1, 10\}$.

For zero-order, we tried $\delta \in \{0.1, 0.01, 0.001\}$.

## F  INTERPRETATION OF THE LEARNED POLICIES

We visually inspected the model's performance on large battles. On the larger Marines map (m15v16), DQN learned to focus fire. Because this map has many units, focus firing leads to units bumping into each other to try to focus on a single unit. The PG player seemed to have a policy that attacks the closest marine, though it doesn't do a good job switching targets. The Marines that are not in range often bump into each other. Our zero order optimization learns a hybrid between focus firing

---

[3]Because several actions are durative, including moves. Moves have a dynamic consisting of per-unit-type turn rate, max speed, and acceleration parameters.

and attacking the closest unit. Units would switch to other units in range if possible, but still focus on specific targets. This leads to most Marines attacking constantly, as well as focus firing when they can. However, the learned strategy was not perfected, since Marines would still split their fire occasionally when left with few units.

In the Wraiths map (w15v17), the DQN player's strategy was hard to decipher. The most likely explanation is that they tried to attack the closest target, though it is likely the algorithm did not converge to a specific strategy. The PG player learned to focus fire. However, because it only takes 6 Wraiths to kill another, 9 actions are "wasted" during the focus firing (at the beginning of the fight, when all our units are alive). Our zero order player learns that focusing only on one enemy is not good, but it does not learn how many attacks are necessary. This leads to a much higher win rate, but the player still assigns more than 6 Wraiths to an enemy target (maybe for robustness to the loss of one of our units), and occasionally will not focus fire when only a few Wraiths are remaining. This is similar to what the zero order player learned during the Marines scenario.

