# Peer review of "Episodic Exploration for Deep Deterministic Policies for StarCraft Micromanagement"

_ICLR 2017 — accepted_

[Official Review · AnonReviewer1 · rating 8 · confidence 4 · 17 Dec 2016]
**Topically relevant work, likely of significant interest**
originality 2 · recommendation (unofficial) 5

This work introduces some StarCraft micro-management tasks (controlling individual units during a battle). These tasks are difficult for recent DeepRL methods due to high-dimensional, variable action spaces (the action space is the task of each unit, the number of units may vary). In such large action spaces, simple exploration strategies (such as epsilon-greedy) perform poorly.

They introduce a novel algorithm ZO to tackle this problem. This algorithm combines ideas from policy gradient, deep networks trained with backpropagation for state embedding and gradient free optimization. The algorithm is well explained and is compared to some existing baselines. Due to the gradient free optimization providing for much better structured exploration, it performs far better.

This is a well-written paper and a novel algorithm which is applied to a very relevant problem. After the success of DeepRL approaches at learning in large state spaces such as visual environment, there is significant interest in applying RL to more structured state and action spaces. The tasks introduced here are interesting environments for these sorts of problems.

It would be helpful if the authors were able to share the source code / specifications for their tasks, to allow other groups to compare against this work.

I found section 5 (the details of the raw inputs and feature encodings) somewhat difficult to understand. In addition to clarifying, the authors might wish to consider whether they could provide the source code to their algorithm or at least the encoder to allow careful comparisons by other work.

Although discussed, there is no baseline comparison with valued based approaches with attempt to do better exploration by modeling uncertainty (such as Bootstrapped DQN). It would useful to understand how such approaches, which also promise better exploration, compare.

It would also be interesting to discuss whether action embedding models such as energy-based approaches (e.g.

[Official Review · AnonReviewer2 · rating 7 · confidence 4 · 19 Dec 2016 (modified: 23 Jan 2017)]
**Final Review: Nice new application of zeroth order optimization for structured exploration. Complex domain, and good results**
soundness 5 · clarity 5 · impact 5 · meaningful comparison 5

The paper presents a learning algorithm for micromanagement of battle scenarios in real-time strategy games. It focuses on a complex sub-problem of the full RTS problem. The assumptions and restrictions made (greedy MDP, distance-based action encoding, etc.) are clear and make sense for this problem.

The main contribution of this paper is the zero-order optimization algorithm and how it is used for structured exploration. This is a nice new application of zero-order optimization meets deep learning for RL, quite well-motivated using similar arguments as DPG. The results show clear wins over vanilla Q-learning and REINFORCE, which is not hard to believe. Although RTS is a very interesting and challenging domain (certainly worthy as a domain of focused research!), it would have been nice to see results on other domains, mainly because it seems that this algorithm could be more generally applicable than just RTS games. Also, evaluation on such a complex domain makes it difficult to predict what other kinds of domains would benefit from this zero-order approach. Maybe the authors could add some text to clarify/motivate this.

There are a few seemingly arbitrary choices that are justified only by "it worked in practice". For example, using only the sign of w / Psi_{theta}(s^k, a^k). Again later: "Also we neglected the argmax operation that chooses the actions". I suppose this and dividing by t could keep things nicely within or close to [-1,1] ? It might make sense to try truncating/normalizing w/Psi; it seems that much information must be lost when only taking the sign. Also lines such as "We did not extensively experiment with the structure of the network, but we found the maxpooling and tanh nonlinearity to be particularly important" and claiming the importance of adagrad over RMSprop without elaboration or providing any details feels somewhat unsatisfactory and leaves the reader wondering why.. e.g. could these only be true in the RTS setup in this paper?

The presentation of the paper can be improved, as some ideas are presented without any context making it unnecessarily confusing. For example, when defining f(\tilde{s}, c) at the top of page 5, the w vector is not explained at all, so the reader is left wondering where it comes from or what its use is. This is explained later, of course, but one sentence on its role here would help contextualize its purpose (maybe refer later to the section where it is described fully). Also page 7: "because we neglected that a single u is sampled for an entire episode"; actually, no, you did mention this in the text above and it's clear from the pseudo-code too.

"perturbated" -> "perturbed"

--- After response period: 

No rebuttal entered, therefore review remains unchanged.

[Official Review · AnonReviewer3 · rating 7 · confidence 4 · 19 Dec 2016]
**No Title**
impact 3 · recommendation (unofficial) 4

This is a very interesting and timely paper, with multiple contributions. 
- it proposes a setup for dealing with combinatorial perception and action-spaces that generalizes to an arbitrary number of units and opponent units,
- it establishes some deep RL baseline results on a collection of Starcraft subdomains,
- it proposes a new algorithm that is a hybrid between black-box optimization REINFORCE, and which facilitates consistent exploration.


As mentioned in an earlier comment, I don’t see why the “gradient of the average cumulative reward” is a reasonable choice, as compared to just the average reward? This over-weights late rewards at the expense of early ones, so the updates are not matching the measured objective. The authors state that they “did not observe a large difference in preliminary experiments” -- so if that is the case, then why not choose the correct objective?

DPQ is characterized incorrectly: despite its name, it does not “collect traces by following deterministic policies”, instead it follows a stochastic behavior policy and learns off-policy about the deterministic policy. Please revise this. 

Gradient-free optimization is also characterized incorrectly (“it only scales to few parameters”), recent work has shown that this can be overcome (e.g. the TORCS paper by Koutnik et al, 2013). This also suggests that your “preliminary experiments with direct exploration in the parameter space” may not have followed best practices in neuroevolution? Did you try out some of the recent variants of NEAT for example, which have been applied to similar domains in the past?

On the specific results, I’m wondering about the DQN transfer from m15v16 to m5v5, obtaining the best win rate of 96% in transfer, despite only reaching 13% (the worst) on the training domain? Is this a typo, or how can you explain that?

[Final Decision · Program Chairs · 06 Feb 2017]
**ICLR committee final decision**

The paper presents an approach to structured exploration in StarCraft micromanagement policies (essentially small tasks in the game). From an application standpoint, this is a notable advance, since the authors are tackling a challenging domain and in doing so develop a novel exploration algorithm that seems to work quite well here.
 
 The main downside of the paper is that the proposed algorithm does seem fairly ad-hoc: certain approximations in the algorithm, like ignoring the argmax term in computing the backprop, are not really justified except in that it "seems to work well in practice" (a quote from the paper), so it's really unclear whether these represent general techniques or just a method that happens to work well on the target domain for poorly understood reasons.
 
 Despite this, however, I think the strength of the application is sufficient here. Deep learning has been alternatively pushed forward by more algorithmic/mathematical advances and more applied advances, with many of the major breakthroughs coming from seemingly ad-hoc strategies applied to challenging problems. This paper falls in that later category: the ZO algorithm may or may not lead to something slightly more disciplined in the future, but for now the compelling results on StarCraft are I believe enough to warrant accepting the paper.
 
 Pros:
 + Substantial performance improvement (over basic techniques like Q-learning) on a challenging task
 + Nice intuitive justification of a new exploration approach
 
 Cons:
 - Proposed algorithm seems rather ad-hoc, making some (admittedly not theoretically justified) approximations simply because they seem to work well in practice